# Fast Random Fourier Features Through Randomized Block-Diagonal Projection

## Abstract

Sparsity has often been used to address expensive matrix multiplication with very large projection matrices for random Fourier features. Traditional sparse projections, however, lead to information loss when sparsity is moderate. In this work, we propose Randomized Block-Diagonal Projection (RBDP) with structured sparsity in a block-diagonal projection matrix and feature shuffling to retain all information of the original features after projection in projected space with high sparsity and efficiency due to the proposed block-diagonal matrix for projection. Error bounds are given in our analysis for the proposed structured sparsity and feature shuffling. We show that our estimators for kernel approximations and random projection are unbiased with the variance inversely proportional to $k$. The proposed method allows much reduced computation with improved complexity $O(\max\{k, D\}n)$, the dimensionality of the feature vector $D$, the dimensionality in the projected space $k$ and the sample size $n$, compared to the complexity $O(kDn)$ where $k \gtrsim 1000$ to keep errors acceptable traditionally associated with random Fourier features and random projection. It is demonstrated in our experiments that the proposed method achieves significant speed improvements, i.e. a speed-up up to 10,000x over Random Kitchen Sinks and a speed-up up to 15x over Fastfood (Le et al., 2013) on real-world datasets. RBDP is a general framework, simple to implement without reliance on the fast Walsh-Hadamard transform, for any shift-invariant kernels with no assumption on the use of Gaussians in the projection matrix. Our code is made available at https://anonymous[1].

## 1 Introduction

Both random Fourier features and random projection are popular methods in classification and regression tasks Bingham & Mannila (2001); Ailon & Chazelle (2006); Anand et al. (2012); Paul et al. (2013); Zhang et al. (2014). Random projection is an efficient and distance-preserving technique while random Fourier features allow non-linear feature mapping through randomization. Random Fourier features, which is closely related to random projection, became popular for good approximations to shift invariant kernels and random Fourier features can be considered as nonlinear random projection Rahimi & Recht (2008). In large-scale real-world problems, the original dimensionality, $D$, the dimensionality in the projected space, $k$, and the sample size, $n$, can be very large. With $k \sim 10^5$ for high accuracies, $D$ from $10^5$ to $10^7$ in Zhai et al. (2014) and $n$ from $10^6$ to $10^7$ in Deng et al. (2009), matrix multiplication required can be prohibitively expensive with the complexity $O(kDn)$.

For random projections, we have $n$ data points $\{\boldsymbol{u}_i\}_{i=1}^n \in \mathbb{R}^D$ in data matrix $\boldsymbol{A} \in \mathbb{R}^{D \times n}$ with $D$ dimensions and a random matrix $\boldsymbol{R} \in \mathbb{R}^{k \times D}$ for projection. For projected data points $\boldsymbol{RA}$, each point $\{\boldsymbol{v}_i\}_{i=1}^n \in \mathbb{R}^k$ is in $k$ dimensions. The computational complexity of traditional random projection is $O(kDn)$ and it is computationally expensive for large-scale problems. It can be easily shown that, as in (Vempala, 2004) and (Li et al., 2006a), we have the expectation for the squared $L^2$-norm of the projected vector $\boldsymbol{v}$ from the original vector $\boldsymbol{u}$ before random projection:

---

[1]Due to anonymity for the review, the link to the code repository will be provided after the review process.

$$\mathbb{E}(\|\boldsymbol{v}_1\|^2) = \|\boldsymbol{u}_1\|^2 = \sum_{j=1}^{D}(\boldsymbol{u}_1)_j^2.$$

Similarly, we get

$$\mathbb{E}(\|\boldsymbol{v}_1 - \boldsymbol{v}_2\|^2) = \|\boldsymbol{u}_1 - \boldsymbol{u}_2\|^2.$$

In addition, $\frac{(\boldsymbol{v}_1)_i}{\sqrt{\|\boldsymbol{u}_1\|^2/k}}$ and $\frac{(\boldsymbol{v}_1)_i - (\boldsymbol{v}_2)_i}{\sqrt{\|\boldsymbol{u}_1 - \boldsymbol{u}_2\|^2/k}}$ both follow the normal distribution meaning that

$$\frac{(\boldsymbol{v}_1)_i}{\sqrt{\|\boldsymbol{u}_1\|^2/k}} \sim \mathcal{N}(0,1),$$

$$\frac{(\boldsymbol{v}_1)_i - (\boldsymbol{v}_2)_i}{\sqrt{\|\boldsymbol{u}_1 - \boldsymbol{u}_2\|^2/k}} \sim \mathcal{N}(0,1). \tag{1}$$

Thus, when we take the sum over all $i$ of $(\boldsymbol{v}_1)_i^2$, we have $\sum_i (\boldsymbol{v}_1)_i^2$ following the $\chi^2$-distribution, i.e.

$$\frac{\|\boldsymbol{v}_1\|^2}{\|\boldsymbol{u}_1\|^2/k} \sim \chi_k^2$$

and, for $\sum_i((\boldsymbol{v}_1)_i - (\boldsymbol{v}_2)_i)^2$,

$$\frac{\|\boldsymbol{v}_1 - \boldsymbol{v}_2\|^2}{\|\boldsymbol{u}_1 - \boldsymbol{u}_2\|^2/k} \sim \chi_k^2 \tag{2}$$

where $\chi_k^2$ denotes the chi-squared random variable with $k$ degrees of freedom. With one of the tightest bounds for the Johnson and Lindenstrauss (JL) lemma in (Achlioptas, 2003b), it is shown that

$$(1 - \epsilon)\|\boldsymbol{u}_1 - \boldsymbol{u}_2\|^2 \le \|\boldsymbol{v}_1 - \boldsymbol{v}_2\|^2 \le (1 + \epsilon)\|\boldsymbol{u}_1 - \boldsymbol{u}_2\|^2$$

with probability $1 - n^{-\gamma}$ given that

$$k \le k_0 = \frac{4 + 2\gamma}{\epsilon^2/2 - \epsilon^3/3}\log(n).$$

For kernel methods with dot products, it can also be shown that, as in (Li et al., 2006a) and (Li et al., 2006b),

$$\mathbb{E}(\boldsymbol{v}_1^T \boldsymbol{v}_2) = \boldsymbol{u}_1^T \boldsymbol{u}_2 = \sum_{j=1}^{D}(\boldsymbol{u}_1)_j(\boldsymbol{u}_2)_j.$$

## 1.1 Kernel Approximation

In this section, we describe how the dot products of vectors with random Fourier features can approximate kernels. For a properly scaled shift-invariant kernel $K(\delta)$, Bochner's theorem guarantees that its Fourier transform $p(\omega)$ is a probability density function (Rahimi & Recht, 2008). It can be shown that

$$\begin{aligned} K(x - y) &= \int_{\mathbb{R}^d} p(w)(\cos(w^T x)\cos(w^T y) + \sin(w^T x)\sin(w^T y)) \\ &= E_p[< (\cos(w^T x), \sin(w^T y)), (\cos(w^T y), \sin(w^T x)) >]. \end{aligned} \tag{3}$$

For $x \in \mathbb{R}^d$, $K(.)$ can be approximated with inner product $< \phi(x), \phi(y) >$. Thus,

$$\phi(x) = \sqrt{\frac{2}{k}}(\cos(w_1^T x), \sin(w_1^T x), \cos(w_2^T x), \sin(w_2^T x), \dots, \cos(w_{k/2}^T x), \sin(w_{k/2}^T x))$$

or, alternatively,

$$\phi(x) = \sqrt{\frac{2}{k}} (\cos(w_1^T x), \cos(w_2^T x), \dots, \cos(w_{k/2}^T x), \sin(w_1^T x), \sin(w_2^T x), \dots, \sin(w_{k/2}^T x))$$

as the dot product $\phi(x)^T \phi(y)$ gives us the same value in the alternate form where $w_1, \dots, w_k$ are drawn according to $p(w)$, i.e.

$$\phi(x)^T \phi(y) = \frac{1}{k/2} \sum_{i=1}^{k/2} \cos(\omega_i^T(x - y)). \tag{4}$$

Our method takes advantage of structured sparsity in a block-diagonal matrix for projection and feature shuffling to retain all information of the original features with high sparsity and efficiency. Feature shuffling is used to reduce correlations between features because the order of features does not affect the distances or similarities between feature vectors, provided that the same order is consistently applied to all feature vectors. Instead of generating evenly spread random non-zero entries with previous methods like Fastfood which uses Walsh Hadamard Transform (WHT), a random order of features in the data matrix is chosen before projection in our method. It is easy to implement the proposed method without the need for libraries for sparse matrix computation or fast WHT which depends on its implementation and the software and hardware architecture. Moreover, as no Gaussian assumption has been made to the random entries of the projection matrix, the proposed method is a unified approach to efficient random projections and random Fourier features with any shift-invariant kernels.

## 1.2 The Complexity

As non-zero elements are arranged in a block-diagonal matrix such that the computational complexity is independent of $k$, the complexity becomes $O(Dn)$ when $k \leq D$, i.e. the number of non-zero elements in the random matrix is reduced to $D$. The proposed method speeds up traditional random projection and Random Kitchen Sinks from $O(kDn)$ to $O(\max\{k, D\}n)$ because the complexity of the proposed method is $O(kn)$ when the dimensionality after projection $k_{multi}$ is larger than $D$. The computational complexity of Random Kitchen Sinks (Rahimi & Recht, 2008) and Fastfood (Le et al., 2013), two state-of-the-art efficient methods for kernel approximations, are $O(kDn)$ and $O(k \log(D)n)$ respectively. Fastfood is a log-linear time algorithm with $O(k \log(D)n)$. Note also that the big-O notation obtained through asymptotic analysis does not account for constant factors due to the fact that when $n$ gets large enough, constants do not matter. Moreover, it has been suggested that the computational speeds of other related methods like Fastfood relying on the Walsh-Hadamard Transform could be impacted by large big-O constants Iwen et al. (2007), Li et al. (2014).

In our experiments, it is demonstrated that there is a bigger computational advantage with the complexity our linear-time method when both $D$ and $k$ are large shown in Table 3, i.e. there is a 15x speed-up for our method over Fastfood on real-world datasets with larger $D$ and $k$ while the speed-up is only around 3x compared to that of Fastfood with small $D$ and small $k$. For the value of $k$, it is shown that empirically, for low errors with random projections and kernel approximations using random Fourier features, $k$ should be larger than or close to $10^4$ (Le et al. (2013), Jacot et al. (2018a), Sutherland & Schneider (2015) Zhang et al. (2019), Nabil (2017)).

## 1.3 Contributions

The complexity of Random Kitchen Sinks (RKS) Rahimi & Recht (2008) and that of a more recent state-of-the-art log-linear time method, Fastfood (Le et al., 2013), are respectively $O(kDn)$ and $O(k \log(D)n)$ where usually $k \gtrsim 1000$ is used to achieve good kernel approximations with random Fourier features and random projection with acceptable errors Jacot et al. (2018b), Zhang et al. (2018), Nabil (2017), Sutherland & Schneider (2015), Le et al. (2014).

The proposed method is a linear-time method. The algorithm is 1.) easy to implement, 2.) with complexity $O(\max\{k, D\}n)$, 3.) designed to provide an unbiased estimator having variance inversely proportional to $k$

and 4.) shown to achieve a speed-up up to 10,000x over Random Kitchen Sinks and a speed-up up to 15x over Fastfood on real-world datasets for the realistic values of $D$ and $k$.

## 2 Related Work

### 2.1 Previous Approaches to Fast Random Projection and Fast Random Fourier Features

Speeding up computation with a sparse random projection matrix with one-third non-zero entries in the matrix is proposed by Achlioptas Achlioptas (2003a). However, with a sparse projection matrix, some features in feature vectors can be totally ignored in computation. Another idea is to make the sparse entries spread more evenly. To do this, one can use the Fast Johnson-Lindenstrauss Transform (FJLT) $\Pi = \mathbf{PHD}$ where $\mathbf{P}$ is the sparse projection matrix with $\boldsymbol{D}$ as a diagonal matrix with $\boldsymbol{D}_{i,i} \in \{+1, -1\}$. Each entry in $\boldsymbol{D}_{i,i}$ is an i.i.d. random variable and $\mathbf{H}$ is the $D \times D$ Walsh Hadamard Transform matrix. The complexity to multiply $\boldsymbol{A}$ by the "mixing matrix" preconditioner $\mathbf{HD}$ matrix with the fast Walsh-Hadamard is $O(D \log D)$. It can be shown that $\mathbf{HD}$ is $L2$-norm preserving to make it a reasonable mixing matrix. In a more efficient method called Improved Subsampled Randomized Hadamard Transform (SRHT) in Boutsidis & Gittens (2012), a subsampling matrix $S$ is considered instead of $P$, i.e. $\Pi = \mathbf{SHD}$ with complexity $O(D \log k)$, original dimensionality $D$ and reduced dimensionality $k$ where $k < D$. For random Fourier features, Fastfood Le et al. (2013) computes random Fourier features which extends previous work with SRHT, sparse JLT and FJLT. Our method with sparse data matrix can theoretically achieve $O(nnz(A))$ with a sparse data matrix and $O(Dn)$ with dense data matrix.

### 2.2 Platform-Specific Implementations of Previous Methods

All implementations for fast WHT and Fastfood we have found with $\mathbf{HD}$ relies on the library SPIRAL[2] introduced in Püschel et al. (2004). In addition, the speed in practice very much depends on the implementation of fast WHT and the computation of sparse matrices. For example, in MATLAB, the simplest way to implement fast WHT or Fastfood is to consider the transform as matrix computation with dense matrices to perform the matrix multiplication which does not take advantage of efficient computation on entries with zeros. This easy-to-implement method however is not very efficient with complexity $O(D^2)$ to compute $\mathbf{HDA}$ and it requires $\Omega(D^2)$ memory. Fortunately, with fast WHT formulated as FFT, there are efficient methods for the transform to take only $O(D \log D)$ instead of direct multiplication with dense matrices.

Although MATLAB comes with a native implementation for the fast WHT, it has been showed empirically in many previous studies that the time required is in reality longer than direct multiplication with the Hadamard matrix. That means there is no speed up with fast WHT in Matlab with the WHT as the bottleneck in the overall computation. This is the reason why many implementations if not all rely on SPIRAL to speed up WHT. SPIRAL written in C as a signal processing package provides an efficient implementation of WHT, to take advantage of specific machine architectures. In a lot of implementations of WHT, SRHT or Fastfood, SPIRAL with mex in Matlab is used for fast WHT and fast multiplication with sparse $\mathbf{P}$ or $\mathbf{S}$. However, efficient multiplications for sparse matrices are platform-dependent Kunchum et al. (2017), Dalton et al. (2015), Liu & Vinter (2014) and Yang et al. (2011).

### 2.3 Other Approaches

Although random projection is computationally more efficient compared to many other dimensionality reduction methods such as principal component analysis (PCA), it is still computationally expensive for very large-scale problems. Methods with sparse random matrices have been proposed to speed up traditional random projection. The method in (Achlioptas, 2003b) with sparse random projection can achieve about a three-fold speed-up compared to vanilla random projection with a small loss of accuracy and (Li et al., 2006a) gets a more efficient $\sqrt{D}$-fold speed-up where $D$ is dimensionality of the input space.

More recently, a very related technique called random Fourier features to speed up kernel methods has attracted a lot of attention. Although the performance of non-linear kernel methods is almost always better

---

[2]https://github.com/jeffeverett/spiral-wht

than that of linear kernel methods, non-linear kernel methods with large-scale problems are known to be prohibitively expensive as they do not scale well with the sample sizes of the training sets. Approximations with non-linear kernel methods aim to reduce time complexity so large-scale non-linear kernel methods can become practical. There are two popular methods for these approximations: 1.) the Nystrom approximation method for Gram matrices (Williams & Seeger, 2001) can be used to speed up general non-linear methods to $\mathcal{O}(nD)$, where $D$ is dimensionality of the input space and $n$ is the number of training examples (Drineas & Mahoney, 2005; Li et al., 2015; Jin et al., 2011). 2.) alternatively, a method called random Fourier features (Rahimi & Recht, 2008) is proposed to approximate non-linear kernels. In this method, the original high-dimensional data is projected to another feature space like random projection. Experiments show that random Fourier features can perform very well with non-linear kernel methods in large-scale classification and regression tasks. Random Fourier features can be used to speed up non-linear kernel methods but the generation of random Fourier features can also be more efficient with a recent method called Fastfood (Le et al., 2013). Experiments for Fastfood show that classification performance with the Nystrom method, original random Fourier features and Fastfood are close while Fastfood is faster than the other two methods. Although the computational efficiency of recent methods for both RP and kernel approximations has been improved, they are still prohibitively expensive when the projected feature space is very large. This is the case especially for random Fourier features. In this paper, an efficient method is proposed for random projections and random Fourier features with the computational complexity independent of $k$.

## 3 Our Method

In this work, we introduce structured sparsity to the projection matrix instead of, randomized sparsity, to retain all information from original features leaving only $D$ non-zero elements in the $k \times D$ projection matrix with sparsity $s = 1/k$ which is the fraction of the number of non-zero random numbers generated in the projection matrix. Together with normalization and the shuffling of features, we found that computing random Fourier features and random projections can be highly efficient. Theoretical analysis is provided for the error with encouraging experimental supports.

For the random matrix of random projection $\boldsymbol{R}$, we create a deterministically sparse matrix $\boldsymbol{S} \in \mathbb{R}^{k \times D}$ with $D(k-1)$ zeros, i.e. only $D$ Gaussian random numbers need to be generated. We show that $\mathbb{E}(\boldsymbol{v}_i) = \boldsymbol{S}\boldsymbol{u}_i$ instead of $\mathbb{E}(\boldsymbol{v}_i) = \frac{1}{\sqrt{k}}\boldsymbol{R}\boldsymbol{u}_i$ from standard random projections. We define $\boldsymbol{S} =$

$$
\begin{pmatrix}
\mathbf{r_1} & 0\ldots0 & 0\ldots0 & 0\ldots0 \\
0\ldots0 & \mathbf{r_2} & 0\ldots0 & 0\ldots0 \\
0\ldots0 & 0\ldots0 & \ddots & 0\ldots0 \\
0\ldots0 & 0\ldots0 & 0\ldots0 & \mathbf{r_k}
\end{pmatrix}
\tag{5}
$$

with each row vector $\{\mathbf{r_i}\}_{i=1}^{k} \in \mathbb{R}^{(D/k)}$. We increase sparsity for more efficient computation to the extent that we can retain all information of the original features in the projected vectors. During projection, each dot product between each row of the projection matrix and the feature vector retains the information of a subset of original features. The resultant $k$-dimensional vector retains all information with the $k$ dot products involving all original features after projection.

### 3.1 The Algorithms

An equivalent formulation of this to find $\boldsymbol{v}_i$ is to calculate the diagonal elements of the matrix $\boldsymbol{C}\boldsymbol{U}$ where we have vector $\boldsymbol{u}_i$ reshaped as $\boldsymbol{U}_i \in \mathbb{R}^{(D/k) \times k}$ and $\boldsymbol{C} = [r_1; r_2; \ldots; r_k] \in \mathbb{R}^{k \times (D/k)}$. Now, for each data point $i$ with $\boldsymbol{U}_i$, we calculate the diagonal elements $diag(\boldsymbol{C}\boldsymbol{U}_i)$ that gives $k$ features in the subspace after random projection, i.e. $\sum_l (r_m)_l \times (\boldsymbol{U}_i)_{l,m}$ gives the element $m$ of the diagonal matrix.

In this section, there are two algorithms. As described in Sub-section 3.2, we first pre-process the data by randomly shuffling the order of the features in each feature vector and normalizing the feature vectors. Our method to speed up the computation for random projections is in Algorithm 1 and for random Fourier features with the Gaussian kernel is in Algorithm 2 with $\boldsymbol{C} = \boldsymbol{C}_G$ generated from the standard normal

distribution for each element. For other kernels, other distributions are required for general $C$ as described in Sub-section 4.1.

---

**Algorithm 1** Fast Random Projection to compute $diag(\boldsymbol{CU})$ with $\boldsymbol{C} = \boldsymbol{C}_G$

> **Input:** $k$ and all data points $\{\boldsymbol{u}_i\}_{i=1}^n \in \mathbb{R}^D$
> **Output:** $\{\boldsymbol{v}_i\}_{i=1}^n \in \mathbb{R}^D$
> **for** $i := 1; n$ **do**
>   $\boldsymbol{v}_i := diag(\boldsymbol{CU}_i)$
> **end for**

---

**Algorithm 2** Fast Fourier Features for Kernel Approximations

> **Input:** $k$ and all data points $\{\boldsymbol{u}_i\}_{i=1}^n \in \mathbb{R}^D$
> **Output:** $\{\boldsymbol{v}_i\}_{i=1}^n \in \mathbb{R}^D$
> **for** $i := 1; n$ **do**
>   $\boldsymbol{v}_i := diag(\sigma \boldsymbol{C}_G \boldsymbol{U}_i)$ for the Gaussian kernel or $\boldsymbol{v}_i := diag(\boldsymbol{CU}_i)$ for other kernels
>   $\boldsymbol{v}_i := \sqrt{k}\boldsymbol{v}_i$
>   $\boldsymbol{v}_i := [\cos(\boldsymbol{v}_i); \sin(\boldsymbol{v}_i)]$
>   $\boldsymbol{v}_i := \frac{1}{\sqrt{k}}\boldsymbol{v}_i$
> **end for**

---

Notice that, with Algorithm 2, the number of random Fourier features generated cannot be more than the original dimensionality, i.e. $k \leq D$. For a larger number of random Fourier features than D, Algorithm 2 is invoked multiple times, i.e. $N_{multi}$ times, to obtain the projected vector in the desired dimensionality after projection $k_{multi} = kN_{multi}$. The sparsity as described previously in Section 3 is $s = 1/k$. With Algorithm 2 invoked multiple times, the sparsity is still $s_{multi} = 1/k$, not $s_{multi} = 1/k_{multi}$.

### 3.2 Random Permutation of Features and Normalization

Motivated by the fact that the chi-squared random variable can be asymptotically approximated by the normal random variable, we consider random permutations of features and feature normalization to speed up random Fourier features.

As shown in Lemma 4.6 in the next section, the bounds for $\|\boldsymbol{v}_1 - \boldsymbol{v}_2\|^2$ depends on the data points $\{\boldsymbol{u}_i\}_{i=1}^n$. When $M/m = 1$ with $m = \min\{\sqrt{\sum_l(\boldsymbol{U})_{l,1}^2}, \sqrt{\sum_l(\boldsymbol{U})_{l,2}^2}, \ldots, \sqrt{\sum_l(\boldsymbol{U})_{l,k}^2}\}$ and $M = \max\{\sqrt{\sum_l(\boldsymbol{U})_{l,1}^2}, \sqrt{\sum_l(\boldsymbol{U})_{l,2}^2}, \ldots, \sqrt{\sum_l(\boldsymbol{U})_{l,k}^2}\}$, we have the tightest bounds, i.e. the inequality reduces back to the original JL lemma but obviously there is no way that we can change the data. We use two techniques which include feature shuffling and normalization to obtain $diag(\boldsymbol{CU})$ so that $M/m$ is small. We will demonstrate that with feature shuffling and feature normalization, $M/m$ is not far from 1.

For data point $i$, we obtain a random permutation of features $(\alpha((\boldsymbol{u}_i)_1), \alpha((\boldsymbol{u}_i)_2), \ldots, \alpha((\boldsymbol{u}_i)_D))$. If we permute the order of the features, $\|\boldsymbol{u}_1 - \boldsymbol{u}_2\|^2$ gives us the same Euclidean distance regardless of the permutation. However, the permutation makes $M/m$ a much closer value to 1 for $\|diag(\boldsymbol{C}(\boldsymbol{U}_1 - \boldsymbol{U}_2))\|^2$ because of less correlations among features after shuffling in $\{(\boldsymbol{U}_i)_{l,m}\}_{l=1}^{(n/k)}$.

It is very common is to scale features for various methods to perform well. We normalize features using mean normalization, i.e. $(\boldsymbol{u}_i)_j := \frac{(\boldsymbol{u}_i)_j - \sum_i(\boldsymbol{u}_i)_j/n}{\max_i(\boldsymbol{u}_i)_j - \min_i(\boldsymbol{u}_i)_j}$ $\quad \forall i, j$.

## 4    Results

The expectation of the approximate kernel in Sutherland & Schneider (2015) with feature vectors $\boldsymbol{u}_1$ and $\boldsymbol{u}_2$ is

$$E_\omega \phi(\boldsymbol{u}_1)^T \phi(\boldsymbol{u}_2) = E_\omega \Big[\frac{1}{k/2} \sum_{i=1}^{k/2} \cos(\omega_i^T(\boldsymbol{u}_1 - \boldsymbol{u}_2))\Big] = E_\omega \Big[\frac{1}{k/2} \sum_{i=1}^{k/2} \cos(\omega_i^T(\Delta_{\boldsymbol{u}}))\Big] \tag{6}$$

where $\Delta_{\boldsymbol{u}} = \boldsymbol{u}_1 - \boldsymbol{u}_2$.

In our analysis, intuitively two cases can be considered. First, for fixed $\Delta_{\boldsymbol{u}}$ and Gaussian $\omega$, with the normal random variable $X = \omega^T \Delta_{\boldsymbol{u}} \sim \mathcal{N}(0, \sigma_x^2)$, it can be easily found that, the expectation is

$$E[\cos(X)] = e^{-\sigma_x^2/2}$$

which is the approximate Gaussian kernel using random Fourier features.

With our method and $\Delta_i = (\boldsymbol{U}_1)_i - (\boldsymbol{U}_2)_i$,

$$E_{\omega,\Delta} \phi(\boldsymbol{u}_1)^T \phi(\boldsymbol{u}_2) = E_{\omega,\Delta}\Big[\frac{1}{k/2} \sum_{i=1}^{k/2} \cos(\omega_i^T \Delta_i)\Big] = \frac{1}{k/2} \sum_{i=1}^{k/2} E_{\omega_i,\Delta_i}[\cos(\omega_i^T \Delta_i)] \tag{7}$$

.

For the second case, with fixed $\omega$ and the pre-processing techniques used in Section 3.2, $E_{\Delta_i}\|\sqrt{k}\Delta_i\|_2^2 = \|\Delta_{\boldsymbol{u}}\|_2^2$, i.e. $\|\Delta_i\|_2^2$ is asymptotically normal. We therefore consider the approximation with error analysis given in Theorem 4.1

$$\|\Delta_{\boldsymbol{u}}\|_2^2 \approx \|\sqrt{k}\Delta_{i \in [1,k/2]}\|_2^2 \sim \mathcal{N}(\mu_{\Delta^2}, \sigma_{\Delta^2}^2) \tag{8}$$

and let $\|\Delta\|_2^2 = \|\sqrt{k}\Delta_i\|_2^2$. For each $i \in [1, k/2]$, $\|\Delta_i\|_2^2$ is asymptotically normal due to the central limit theorem and also techniques used in Section 3.2 to make each chi-square normalized and independent. The only assumption here is normality with dependent features justified later in Section 4.2 using feature shuffling, the Shapiro-Wilk test and the central limit theorem for weakly dependent random variables. $E_{\|\Delta\|_2^2 \sim \mathcal{N}(\mu_{\Delta^2}, \sigma_{\Delta^2}^2)}[\cos(\omega^T \Delta)] = E[K(\|\Delta\|_2^2)]$ where $K(\|\Delta\|_2^2)$ becomes $exp(-\gamma X)$ with the Gaussian kernel for example.

For the rest of the analysis, we, formally, bound errors with both $\omega$ and $\Delta$ as random variables using the total expectation and the total variance. We have $E\|\omega_i^T(\sqrt{k}\Delta_i)\|_2^2 = \|\Delta_{\boldsymbol{u}}\|_2^2$ because $\frac{\sqrt{k}\Delta_i}{\sqrt{\|\Delta_{\boldsymbol{u}}\|_2^2}} \sim \mathcal{N}(0, 1)$.

We found, in the analysis, that the expectation of the approximate kernel with our new method using Equation 13 is

$$E_{\omega,\Delta}[\cos(\omega^T \Delta)] = E_\Delta[K(\Delta)] = \sum_i K(\Delta_i)$$

where $\sum_{i=1}^{k/2} K(\Delta_i)$ is the expectation $E_\Delta[K(\Delta)]$ by definition.

**Theorem 4.1.** *With $\|\Delta\|_2^2, \|\sqrt{k}\Delta_i\|_2^2 \sim \mathcal{N}(\mu_{\Delta^2}, \sigma_{\Delta^2}^2)$ following the normal distribution for any $i$, the expectation and the variance for random Fourier features with our method are*

$$E_{\omega,\Delta}\left[\frac{1}{k/2} \sum_{i=1}^{k/2} \cos(\omega_i^T(\sqrt{k}\Delta_i))\right] = E_\Delta[K(\Delta)]$$

$$Var_{\omega,\Delta}\left[\frac{1}{k/2} \sum_{i=1}^{k/2} \cos(\omega_i^T(\sqrt{k}\Delta_i))\right] = \frac{1}{k/2}\left(\frac{1}{2} + \frac{1}{2}E_\Delta[K(2\Delta)] + E_\Delta[K(\Delta)]^2\right)$$

*where the density function $p(\omega_i)$ is the Fourier transform of the kernel $K(\delta)$.*

**Proposition 4.2.** *The unbiased estimator for the Gaussian kernel approximation is*

$$\phi^T(x)\phi(y)\left[\frac{exp(-\gamma\mu_{\Delta^2})}{exp(-\gamma\mu_{\Delta^2} + \gamma^2\sigma_{\Delta^2}^2/2)}\right]$$

*where $\gamma$ is the parameter of the Gaussian kernel $K(\Delta) = exp(-\gamma\|\Delta\|_2^2)$ and the bias correction term is*

$$\frac{exp(-\gamma\mu_{\Delta^2})}{exp(-\gamma\mu_{\Delta^2} + \gamma^2\sigma_{\Delta^2}^2/2)} \approx 1$$

*if $\mu_{\Delta^2}/\sigma_{\Delta^2} >> 1$.*

**Proposition 4.3.** *For the Gaussian kernel $K(\Delta) = exp(-\gamma\|\Delta\|_2^2)$ also called the Radial Basis Function kernel and the squared exponential kernel, using Theorem 4.1 with $\|\Delta\|_2^2, \|\sqrt{k}\Delta_i\|_2^2 \sim \mathcal{N}(\mu_{\Delta^2}, \sigma_{\Delta^2}^2)$ following the normal distribution for any i with the density function $p(\omega_i)$ being the Fourier transform of the kernel $K(\delta)$,*

$$E_{\omega,\Delta}\left[\frac{1}{k/2}\sum_{i=1}^{k/2}\cos(\omega_i^T(\sqrt{k}\Delta_i))\right] = exp(-\gamma\mu_{\Delta^2} + \gamma^2\sigma_{\Delta^2}^2/2),$$

$$Var_{\omega,\Delta}\left[\frac{1}{k/2}\sum_{i=1}^{k/2}\cos(\omega_i^T(\sqrt{k}\Delta_i))\right] = \frac{1}{k/2}\left(\frac{1}{2} + \frac{1}{2}exp(-2\gamma\mu_{\Delta^2} + 2\gamma^2\sigma_{\Delta^2}^2) + exp(-\gamma\mu_{\Delta^2} + \gamma^2\sigma_{\Delta^2}^2/2)^2\right)$$

*where both the expectation and the variance are functions of $\mu_{\Delta^2}$ and $\sigma_{\Delta^2}$.*

**Proposition 4.4.** *For the spherical kernel,*

$$K(\Delta) = 1 - \frac{3}{2}\frac{\|\Delta\|}{\theta} + \frac{1}{2}\left(\frac{\|\Delta\|}{\theta}\right)^3$$

*if $\|\Delta\| < \theta$. 0 otherwise. With $\|\Delta\|_2^2, \|\sqrt{k}\Delta_i\|_2^2 \sim \mathcal{N}(\mu_{\Delta^2}, \sigma_{\Delta^2}^2)$ following the normal distribution for any i, the expectation and the variance for the kernel are respectively*

$$E_{\omega,\Delta}\left[\frac{1}{k/2}\sum_{i=1}^{k/2}\cos(\omega_i^T(\sqrt{k}\Delta_i))\right] = E_{\Delta}[K(\Delta)] = 1 - \frac{3\mu_{\Delta}}{2\theta} + \frac{\mu_{\Delta}^3 + 3\mu_{\Delta}\sigma_{\Delta}^2}{2\theta^3},$$

$$Var_{\omega,\Delta}\left[\frac{1}{k/2}\sum_{i=1}^{k/2}\cos(\omega_i^T(\sqrt{k}\Delta_i))\right] = \frac{1}{k/2}\left(1 - \frac{9\mu_{\Delta}}{2\theta} + \frac{9\mu_{\Delta}^2}{4\theta^2} + \frac{3\mu_{\Delta}^3 + 9\mu_{\Delta}\sigma_{\Delta}^2}{\theta^3} - \frac{3\mu_{\Delta}^4 + 9\mu_{\Delta}^2\sigma_{\Delta}^2}{2\theta^4} + \frac{6\mu_{\Delta}^4\sigma_{\Delta}^2 + \mu_{\Delta}^6 + 9\mu_{\Delta}^2\sigma_{\Delta}^4}{4\theta^6}\right)$$

*where the density function $p(\omega_i)$ is the Fourier transform of the kernel $K(\delta)$.*

**Lemma 4.5.** *With $C \in \mathbb{R}^{k \times (D/k)}$, a random matrix with $k \times (D/k) = D$ elements, and each element following the normal distribution $\mathcal{N}(0,1)$ where $D$ is the original dimensionality and $k$ is the dimensionality after projection, the expectation of $\|v\|^2$ is $\mathbb{E}\{\sum_i^k[diag(CU)]_i^2\} = \|u\|^2$, and, for each element of $v$, $\frac{(v)_j}{\sqrt{\sum_l(U)_{l,j}^2}} \sim \mathcal{N}(0,1)$.*

Note that $\mathbb{E}\{\sum_i^k[diag(CU)]_i^2\} = \|u\|^2$ while we have $\mathbb{E}(\|\frac{1}{\sqrt{k}}Ru\|^2) = \|u\|^2$ for traditional random projection. However, now $\|v\|_2^2$ follows the generalized chi-squared distribution with non-unit variances.

**Lemma 4.6.** *With probability $1 - 2e^{-(\epsilon^2-\epsilon^3)k/4}$,*

$$(1-\epsilon)(m/M)\|u_1 - u_2\|^2 \leq \|v_1 - v_2\|^2 \leq (1+\epsilon)(M/m)\|u_1 - u_2\|^2$$

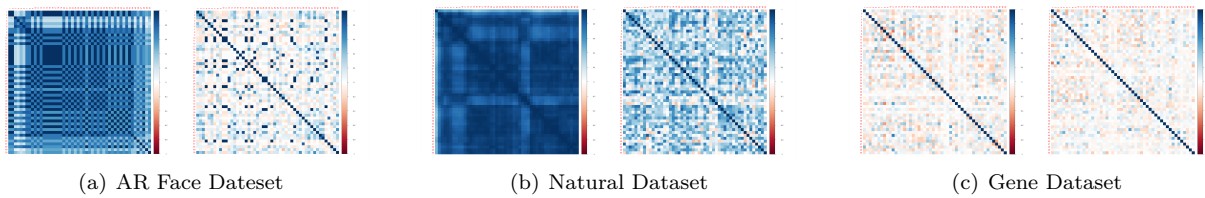

| (a) AR Face Dateset | (b) Natural Dataset | (c) Gene Dataset |

Figure 1: Comparison of feature correlations matrices before and after shuffling. Darker colors denote higher correlations. In a sub-figure, the matrix on the left is obtained before shuffling. Shuffling can significantly reduce the features correlation.

*where* $\|\boldsymbol{u}_1\| = \sum_l \sum_m (\boldsymbol{U}_1)^2_{l,m}$ *and*

$$m = \min\{\sqrt{\sum_l (\boldsymbol{U})^2_{l,1}}, \sqrt{\sum_l (\boldsymbol{U})^2_{l,2}}, \cdots, \sqrt{\sum_l (\boldsymbol{U})^2_{l,k}}\}$$

$$M = \max\{\sqrt{\sum_l (\boldsymbol{U})^2_{l,1}}, \sqrt{\sum_l (\boldsymbol{U})^2_{l,2}}, \cdots, \sqrt{\sum_l (\boldsymbol{U})^2_{l,k}}\}$$

**Theorem 4.7.** *The expectation and the variance for fast random projection with our method are*

$$E_{\omega,\Delta}[\sum_i (\omega_i^T \Delta_i)^2] = \mu_{\Delta^2}, \quad and \quad Var_{\omega,\Delta}[\sum_i (\omega_i^T \Delta_i)^2] = (2\mu_{\Delta^2} + \sigma^2_{\Delta^2})/k$$

*where* $\|\Delta_i\|^2_2 \sim \mathcal{N}(\mu_{\Delta^2}, \sigma^2_{\Delta^2})$ *and* $\omega_i \sim \mathcal{N}(0,1)$.

### 4.1 Other Kernels

$w^T x$ is exactly what we compute for random projections. Thus, we can use the same method to compute $w^T x$ with $diag(\boldsymbol{CU})$ because all elements in $[w_1; w_2; \ldots; w_D]$ follow the normal distribution if we use the Gaussian kernel. Otherwise, other distributions can be used to generate $w$ for other kernels.

### 4.2 The Central Limit Theorem for Weakly Dependent Random Variables

In this sub-section, we first study the effect of the shuffling operation on reducing the correlations between features, the actual speed-ups and the approximation quality using the three datasets which are used for all other experiments as well. Moreover, we investigate whether the value of $\Delta_i$ is normally distributed or close to normality Fleermann & Kirsch (2022), Ermakov & Ostrovskii (1986), Serfling (1968).

The comparison of feature corrections is shown in Figure 1. For all datasets, we randomly pick two examples and evaluate their feature correlation matrices before and after shuffling. To visualize correlations, the correlation matrices with the first 50 features of the examples are shown. Darker colors denote higher correlations. It can be observed the shuffling operation can significantly reduce correlations between features with feature correlation matrices on the left in Sub-figures 1(a) and 1(b) much lighter than those on the right. In Sub-figure 1(c), both matrices are light since the feature correlations for gene expression are relatively low.

The Shapiro-Wilk test is used to examine whether the value of $\Delta_i$ is normal distribution or not. The results from the Shapiro-Wilk test are shown in Table 1 with the dimensionality of projected features $k_{multi}$ (see Section 3.1). On the AR dataset and the natural-image dataset with shuffling, the test suggests that $\Delta_i$ is normal distributed. As the dimensionality of the gene data is only 17,000, when $k_{multi}$ equal to 1,000, there are only 17 elements for the calculation of $\Delta_i$. Hence, in this experiment, we set the values of $k_{multi}$ to 200 and 1,000. In Table 1, with shuffling and normalization, the values of $\Delta_i$ on all three datasets are normally distributed when $k_{multi} = 200$, i.e. the p-value higher than 0.05 and the W value close to 1.

Table 1: Results of Shapiro-Wilk test on three datasets. This test is to verify whether the value of $\Delta_i$ is normal distribution or not.

| Operation | | $k_{multi}$ | AR Dataset | | Natural Dataset | | Gene Dataset | |
|---|---|---|---|---|---|---|---|---|
| Shuff. | Norm. | | W | p-value | W | p-value | W | p-value |
| - | - | 200 | 0.91 | 1.3E-07 | 0.98 | 0.003 | 0.98 | 0.003 |
| | | 1000 | 0.87 | 2.2E-16 | 0.99 | 1.93E-06 | 0.95 | 2.2E-16 |
| $\checkmark$ | - | 200 | 0.99 | 0.552 | 0.99 | 0.571 | 0.98 | 0.010 |
| | | 1000 | 0.99 | 0.590 | 0.99 | 0.800 | 0.95 | 2.2E-16 |
| $\checkmark$ | $\checkmark$ | 200 | 0.99 | 0.582 | 0.99 | 0.820 | 0.99 | 0.783 |
| | | 1000 | 0.99 | 0.544 | 0.99 | 5.12E-01 | 0.99 | 1.89E-06 |
| - | $\checkmark$ | 200 | 0.96 | 1.09E-05 | 0.96 | 0.008 | 0.98 | 0.037 |
| | | 1000 | 0.93 | 2.2E-16 | 0.99 | 1.93E-06 | 0.98 | 2.82E-11 |

## 5 Experiments

In this section, we first demonstrate the speed improvements of the proposed kernel approximation and random projection method. In addition, we conduct a comparative analysis against state-of-the-art techniques to highlight the fact that our method not only speeds up traditional approaches but also preserves comparable approximation quality by assessing the quality of our method with classification and regression tasks. The empirical evidence supports that our approach to kernel approximation allows the linear SVM to reach classification and regression performance on par with that of the non-linear SVM using the radial basis function (RBF) kernel. We implement our method and RKS. They are trained with the same protocol. For Fastfood, we use the code provide on the scikit-learn-extra website[3].

### 5.1 The Actual Speed-up

The real-world time efficiency of the proposed method is evaluated on synthesized datasets and public datasets[4] including the AR face image dataset, a natural image dataset, and a gene dataset. The AR face dataset Martinez & Benavente (1998) contains 3,276 images with 126 people, and the resolution of the images is $576 \times 768$. By following Le et al. (2013); Li et al. (2006a), each image with all pixel values is flattened into a vector. For a grey image of the AR dataset, the dimensionality of its vector is 442,368. Face images in the AR dataset are different from general images because there is always a completely white background in the image. Therefore, a popular natural image dataset Weber (2018) is also used for our evaluation. There are images in three different resolutions in this dataset. To fairly compare with the results on face images, only images in the $512 \times 512$ resolution are chosen in our experiments with this dataset. The images are first converted into gray-scale images meaning that the vectors obtained for the images are 262,144-dimensional. Finally, a biomedical dataset with genes for breast cancer called TCGA (BC-TCGA) Xie et al. (2016) is used to evaluate our method using gene expression bio-sequences. This dataset contains 590 examples with 17,814 genes. All the 590 examples are used in the experiments.

The proposed method is assessed with both random projection and kernel approximation in terms of computational efficiency. The runtime improvement of our method relative to vanilla random projection is shown in Table 2. We make synthesized datasets with various dimensionalities to evaluate the speed improvement of the proposed method. Here, the reduced dimensionality $k_{multi}$ is equal to $k$ which is set from 1,000 to 5,000. When $k = 1,000$, the real-world runtime of our method is 0.31, 0.05, and 0.079 seconds on the three public datasets, while it is 0.023, 0.031, and 0.1 seconds on the synthesized dataset. As the value of $k_{multi} = k$ increases, the running time of vanilla random projection increases quickly, because its complexity is $O(kDn)$. The $k_{multi} = k$ is not a important factor affecting the running time of our method with $O(Dn)$. Hence, the proposed method is faster than vanilla random projection, and the actual speed-up of our method is up to 1226 times.

---

[3]https://scikit-learn-extra.readthedocs.io/en/stable/index.html.
[4]Available at http://www2.ece.ohio-state.edu/aleix/ARdatabase.html/, http://sipi.usc.edu/database/
and https://data.mendeley.com/datasets/ respectively.

Table 2: Speed-up of the proposed method for random projection. Our method speeds up traditional random projection from $O(kDn)$ to $O(Dn)$ when $D$ is larger than the dimensionality after projection $k_{multi} = k$. The actual speed-up of our method is up to 1226 times with $D$ as the dimensionality of input data. $k$ is a hyper-parameter of Algorithm 1.

| Datasets | D | k = 1,000 | k = 2,000 | k = 3,000 | k = 4,000 | k = 5,000 |
|---|---|---|---|---|---|---|
| | D = 5,000 | 23.9x | 40.0x | 65.2x | 74.1x | 100.0x |
| Synth. Dataset | D = 10,000 | 32.3x | 58.8x | 93.9x | 120.6x | 150.0x |
| | D = 100,000 | 106.0x | 193.6x | 316.0x | 405.0x | 462.7x |
| AR dataset | D = 440,000 | 142.9x | 265.9x | 389.1x | 559.7x | 672.0x |
| Nat. dataset | D = 260,000 | 264.0x | 408.3x | 768.0x | 941.6x | 1226.6x |
| Gene dataset | D = 17500 | 67.1x | 123.3x | 203.7x | 241.6x | 331.0 x |

Table 3: For kernel approximation, the proposed method and Fastfood are faster than RKS. The runtime improvements of the two approaches relative to RKS are listed.

| Datasets | D | Methods | k = $10^3$ | k = $5*10^3$ | k = $10^4$ | k = $5*10^4$ | k = $10^5$ | k = $2*10^5$ |
|---|---|---|---|---|---|---|---|---|
| | D = 5,000 | Ours | 8.9x | 45x | 44.6x | 48.3x | 48.8x | 52.8x |
| | | Fastfood | 2.6x | 12x | 13.3x | 17.9x | 20.5x | 21.7x |
| Synthesized | D = 10,000 | Ours | 9.5x | 50.6 x | 95.5 x | 109.3 x | 109.4 x | 109.4 x |
| Dataset | | Fastfood | 2.4x | 10x | 21.8x | 32.6x | 41.3x | 40.3x |
| | D = 100,000 | Ours | 9.2x | 50.6x | 95.2x | 602x | 1300x | 1139x |
| | | Fastfood | 2.7x | 13.6x | 29.3x | 162.1x | 352.8x | 355.2x |
| AR | D = 440,000 | Ours | 13.4x | 72.1x | 153.2x | 767.8x | 1585x | 3361x |
| Dataset | | Fastfood | 2.6x | 12.6x | 28.9x | 141.9x | 267.7x | 554.7x |
| Nat. | D = 260,000 | Ours | 32.3x | 168.5x | 432.6x | 2800x | 4943x | 11716x |
| Dataset | | Fastfood | 3.2x | 16.4x | 32.7x | 175.7x | 345.6x | 794.5x |
| Gene | D = 17500 | Ours | 6.5x | 31.3x | 65.1x | 63.5x | 66.5x | 66.1x |
| Dataset | | Fastfood | 0.6x | 3.0x | 6.1x | 15.7x | 16.1x | 16.0x |

For kernel approximation, the proposed method is compared with other two state-of-the-art kernel approximation approaches, RKS (Rahimi & Recht, 2008) and Fastfood (Le et al., 2013). We vary the parameter $k_{multi} = k$ from $1,000$ to $200,000$ to assess performance disparities. Both our method and Fastfood outperform RKS in speed (see Table 2 detailing the speed-up factors of our method and Fastfood in comparison to RKS). Specifically, when $k = 1,000$, the real-world runtime of our method is respectively 4.6, 0.49, and 0.88 seconds on the three public datasets, while it is 0.062, 0.11, and 1.18 seconds on the synthesized dataset (with three different feature dimensionalities $D$). It is encouraging to see that , when $k_{multi} = k = 200,000$, the speed improvement of our method significantly increases to up to 11,716 times compared to RKS, and it also achieves a 14.7 times speed advantage over Fastfood. These results show that our method is moderately more efficient than Fastfood and significantly more efficient than RKS.

Results in Table 2 and Table 3 demonstrate that our method can significantly improve real-world time efficiency for random projection and kernel approximation. Calculating diag(AB) can be very expensive, in R or Matlab for example, as the product matrix $AB$ is computed first and the diagonal elements are then taken. In our implementation, it is much more efficient to find diag(A*B) with sum(A.*B',2) in Matlab or R.

## 5.2 Approximation Quality

The approximation quality of the proposed method is very close to that of RKS, while our method can significantly improve the runtime (see Section 5.1).

In random projection, approximation quality is measured to see how well the pairwise Euclidean distances among projected vectors can approximate the corresponding distances with the original vectors. The averaged absolute difference between pairwise Euclidean distances before and after projection are used to quantify the approximation quality. These pairwise distances are computed over all example pairs, and the averaged

absolute difference gives us the error. For a dataset with $n$ examples, the average absolute error over all $\binom{n}{2}$ pairs is obtained with

$$Err_{rp} = \frac{1}{\binom{n}{2}} \sum_{i=1}^{n-1} \sum_{j=i+1}^{n} \left| \|\boldsymbol{u}_i - \boldsymbol{u}_j\|^2 - \|\boldsymbol{v}_i - \boldsymbol{v}_j\|^2 \right|, \tag{9}$$

where $\boldsymbol{u}_i$, $\boldsymbol{u}_j$ are two data points, and $\boldsymbol{v}_i$, $\boldsymbol{v}_j$ are the projected points of them. In kernel approximation, approximation quality is quantified using the average absolute error between the approximated kernel values using dot products obtained by the proposed method and the original kernel values over all data point pairs. For a dataset with $n$ examples, the average absolute error over all $\binom{n}{2}$ pairs is given by

$$Err_{rff} = \frac{1}{\binom{n}{2}} \sum_{i=1}^{n-1} \sum_{j=i+1}^{n} \left| \langle \boldsymbol{v}_i, \boldsymbol{v}_j \rangle - K\left(\boldsymbol{u}_i, \boldsymbol{u}_j\right) \right|, \tag{10}$$

where $\langle \boldsymbol{v}_i, \boldsymbol{v}_j \rangle = \boldsymbol{v}_i \cdot \boldsymbol{v}_j$.

For random projection, the left sub-figure of Figure 2 shows approximation error (see Equation 9) against the dimensionality after projection. Comparing with the vanilla method (red), which shows the state-of-the-art approximation quality, the error (y-axis) from our method (green) is very close on all datasets. It indicates

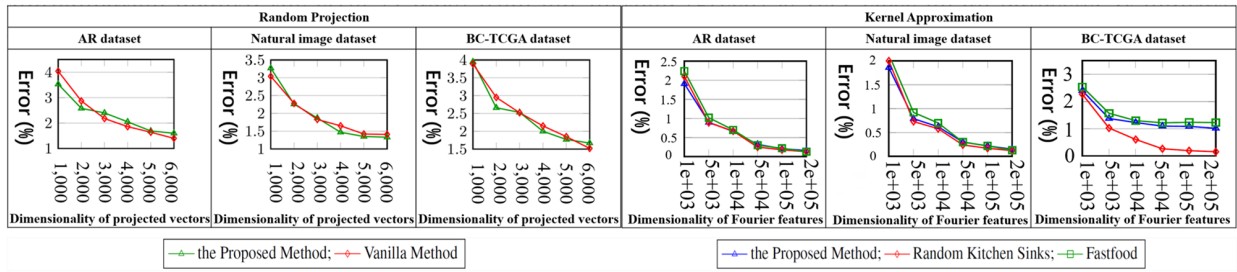

Figure 2: Comparison of approximation error in random projection and kernel approximation. The error is obtained with Equation 9 and Equation 10. It shows the error from the proposed method on the y-axis is close to that from previous methods in all cases. The approximation quality of our method is close to that of previous methods, while the proposed method can significantly speed-up them.

Table 4: Comparison with four SVM variants. Experimental results show that the classification accuracies and regression performance of the linear SVM with our method are very close to the SVM with RBF.

| Datasets | Classification (Accuracy) | | | | | | Regression (RMSE) | | |
|---|---|---|---|---|---|---|---|---|---|
| | ADULT | | | CIFAR-10 | | | CENSUS | | |
| Reduced Dim. | 1000 | 2000 | 3000 | 1000 | 2000 | 3000 | 1000 | 2000 | 3000 |
| Linear SVM (Ours) | 59.4% | 62.2% | 64.5% | 75.9% | 75.8% | 76.3% | 3.1% | 2.9% | 2.8% |
| Linear SVM (RKS) | 58.6% | 62.0% | 63.7% | 75.1% | 76.2% | 76.2% | 2.9% | 2.8% | 2.8% |
| Linear SVM (Fastfood) | 59.1% | 62.2% | 64.3% | 75.1% | 75.6% | 75.7% | 3.1% | 2.7% | 2.8% |
| SVM with RBF | 64.7% | | | 76.3% | | | 1.1% | | |

For kernel approximation, the right sub-figure of Figure 2 shows the error calculated using Equation 10 against the dimensionality of Fourier features. On the AR dataset and the natural dataset, the error from the proposed method (blue) is on par with that of RKS Rahimi & Recht (2008) and Fastfood (Le et al., 2013). The approximation quality of our method is close to that of RKS. In the gene dataset, the error of our method and Fastfood are higher than that of RKS. This is due to the relatively low dimensionality of the gene dataset. Table 1 shown that the $\Delta_i$ is not a normal distribution when $k_{multi} = 1000$. This affects the approximation quality of our method. In Section 5.3, we further investigate the effect of this errors on SVMs for classification and regression.

## 5.3 Performance with the SVM

In this subsection, we further evaluate the actual performance of SVMs with the proposed method. It is found that our approach not only significantly accelerates the speed of traditional methods but also

achieves approximation quality that is comparable to conventional methods. The primary objective of kernel approximation is to enhance the efficiency of kernel method computations without compromising on quality. To this end, we compare the non-linear SVM with the RBF kernel against the linear SVM using three different kernel approximation techniques: the proposed method, RKS(Rahimi & Recht, 2008), and Fastfood(Le et al., 2013). We follow Rahimi & Recht (2008) to apply SVMs to classification and regression on the adult dataset, the census dataset, and the CIFAR-10 dataset [5]. Moreover, the datasets are pre-processed with the same techniques as in Rahimi & Recht (2008).

The classification accuracies and the root-mean-square errors (RMSE) are given with different methods in Table 4. As shown in Table 4, the performance of the linear SVM with the proposed method is on par with that of the non-linear SVM with the RBF. The approximation quality of our proposed method with classification is also found to be encouraging. Our method can significantly speed-up the previous methods.

## 6 Conclusion

In this work, a simple and efficient approach to sparse random projection and efficient computation of random Fourier features is proposed with complexity $O(\max\{k, D\}n)$. The novel method does not rely on specialized libraries for sparse matrix computation or fast WHT. It can be easily implemented. In addition, no Gaussian assumption has been made to the random entries of the projection matrix for random projections and random Fourier features with any shift-invariant kernels with the bias, the variance and error bounds provided. It is shown that the speed-up of our method is up to 10,000 times on real-world datasets compared to RKS and up to 15 times compared to Fastfood (Le et al., 2013).

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

# A    Appendix

## A.1    Analysis for Fast Random Features with Our Method

**Theorem 4.1.** *With $\|\Delta\|_2^2, \|\sqrt{k}\Delta_i\|_2^2 \sim \mathcal{N}(\mu_{\Delta^2}, \sigma_{\Delta^2}^2)$ following the normal distribution for any $i$, the expectation and the variance for random Fourier features with our method are*

$$E_{\omega,\Delta}\left[\frac{1}{k/2}\sum_{i=1}^{k/2}\cos(\omega_i^T(\sqrt{k}\Delta_i))\right] = E_\Delta[K(\Delta)]$$

$$Var_{\omega,\Delta}\left[\frac{1}{k/2}\sum_{i=1}^{k/2}\cos(\omega_i^T(\sqrt{k}\Delta_i))\right] = \frac{1}{k/2}\left(\frac{1}{2} + \frac{1}{2}E_\Delta[K(2\Delta)] + E_\Delta[K(\Delta)]^2\right)$$

*where the density function $p(\omega_i)$ is the Fourier transform of the kernel $K(\delta)$.*

*Proof.* One can obtain, for each pair of data points, $\hat{\mu}$ and $\hat{\sigma}$ using $\cos(\omega^T\Delta)$ (Sutherland & Schneider (2015)) for any given $\Delta$:

$$E[\cos(\omega^T\Delta)] = K(\Delta) \tag{11}$$

$$Var[\cos(\omega^T\Delta)] = \frac{1}{2} + \frac{1}{2}K(2\Delta) - K(\Delta)^2 \tag{12}$$

For the total expectation or the unconditional expectation, $E_Y[E_X[X|Y]] = E_X[X]$ for any two random variables $X$ and $Y$. We have

$$E_{\omega,\Delta}[\cos(\omega^T\Delta)] = E_\Delta[E_\omega[\cos(\omega^T\Delta)|\Delta]] = E_\Delta[K(\Delta)] \tag{13}$$

For the total variance with the law of total variance $Var_Y(Y) = E_X(Var_Y(Y|X)) + Var_X(E_Y(Y|X))$, it is the sum of the expected value of the conditional variance and the variance of the conditional means.

$$Var_{\omega,\Delta}[\cos(\omega^T\Delta)] = E_\Delta[Var_\omega[\cos(\omega^T\Delta)|\Delta]] + Var_\Delta[E_\omega[\cos(\omega^T\Delta)|\Delta]]$$

Using Equation 12 and $Var(X) = E[X^2] - (E[X])^2$,

$$Var_\omega[\cos(\omega^T\Delta)]$$
$$= E_\Delta[\frac{1}{2} + \frac{1}{2}K(2\Delta) - K(\Delta)^2] + Var_\Delta[K(\Delta)]$$
$$= \frac{1}{2} + \frac{1}{2}E_\Delta[K(2\Delta)] - (Var_\Delta[K(\Delta)] - E_\Delta[K(\Delta)]^2) + Var_\Delta[K(\Delta)]$$
$$= \frac{1}{2} + \frac{1}{2}E_\Delta[K(2\Delta)] + E_\Delta[K(\Delta)]^2 \tag{14}$$

As the expectation and the variance of the average, $E[\bar{X}] = \mu_X$ and $Var[\bar{X}] = \frac{1}{n}\sigma_X^2$, with random variables $X_1, X_2, \ldots, X_n$, using Equation 13, we have

$$E_{\omega,\Delta}\left[\frac{1}{k/2}\sum_{i=1}^{k/2}\cos(\omega_i^T\Delta_i)\right] = E_{\omega,\Delta}[\cos(\omega^T\Delta) = E_\Delta[K(\Delta)]$$

and, using Equation 14,

$$Var_{\omega,\Delta}\left[\frac{1}{k/2}\sum_{i=1}^{k/2}\cos(\omega_i^T\Delta_i)\right] = \frac{1}{k/2}Var_\omega[\cos(\omega^T\Delta)]$$

$$= \frac{1}{k/2}\left(\frac{1}{2} + \frac{1}{2}E_\Delta[K(2\Delta)] + E_\Delta[K(\Delta)]^2\right)$$

$\square$

**Proposition 4.2.** *The unbiased estimator for the Gaussian kernel approximation is*

$$\phi^T(x)\phi(y)\left[\frac{exp(-\gamma\mu_{\Delta^2})}{exp(-\gamma\mu_{\Delta^2} + \gamma^2\sigma_{\Delta^2}^2/2)}\right]$$

*where $\gamma$ is the parameter of the Gaussian kernel $K(\Delta) = exp(-\gamma\|\Delta\|_2^2)$ and the bias correction term is*

$$\frac{exp(-\gamma\mu_{\Delta^2})}{exp(-\gamma\mu_{\Delta^2} + \gamma^2\sigma_{\Delta^2}^2/2)} \approx 1$$

*if $\mu_{\Delta^2}/\sigma_{\Delta^2} >> 1$.*

*Proof.* The bias can be found with

$$E_{\omega,\Delta}[\cos(\omega^T\Delta)] = E_\Delta[K(\Delta)]$$

from Theorem 4.1 and

$$E_{X\sim\mathcal{N}}[exp(-\gamma X)] = \sum_i exp(-\gamma\|\Delta_i\|_2^2).$$

$exp(-\gamma\|\Delta_i\|_2^2)$ following the log-normal distribution can be approximated by the normal distribution when $\mu_{\Delta^2}/\sigma_{\Delta^2} >> 1$ and the summation $\sum_i exp(-\gamma\|\Delta_i\|_2^2)$ makes the sum of log-normal random variables follow more closely to the normal distribution due to the central limit theorem. As the expectation of the summation $\sum_i exp(-\gamma\|\Delta_i\|_2^2)$ is just the expectation of the log-normal distribution, we have

$$\sum_i exp(-\gamma\|\Delta_i\|_2^2) = exp(-\gamma\mu_{\Delta^2} + (\gamma\sigma_{\Delta^2})^2/2).$$

The unbiased estimator for our kernel approximation becomes

$$\phi^T(x)\phi(y)[exp(-\gamma\mu_{\Delta^2})/exp(-\gamma\mu_{\Delta^2} + (\gamma\sigma_{\Delta^2})^2/2)].$$

$\square$

**Proposition 4.3.** *For the Gaussian kernel $K(\Delta) = exp(-\gamma\|\Delta\|_2^2)$ also called the Radial Basis Function kernel and the squared exponential kernel, using Theorem 4.1 with $\|\Delta\|_2^2, \|\sqrt{k}\Delta_i\|_2^2 \sim \mathcal{N}(\mu_{\Delta^2}, \sigma_{\Delta^2}^2)$ following the normal distribution for any i with the density function $p(\omega_i)$ being the Fourier transform of the kernel $K(\delta)$,*

$$E_{\omega,\Delta}\left[\frac{1}{k/2}\sum_{i=1}^{k/2}\cos(\omega_i^T(\sqrt{k}\Delta_i))\right] = exp(-\gamma\mu_{\Delta^2} + \gamma^2\sigma_{\Delta^2}^2/2),$$

$$Var_{\omega,\Delta}\left[\frac{1}{k/2}\sum_{i=1}^{k/2}\cos(\omega_i^T(\sqrt{k}\Delta_i))\right] = \frac{1}{k/2}\left(\frac{1}{2} + \frac{1}{2}exp(-2\gamma\mu_{\Delta^2} + 2\gamma^2\sigma_{\Delta^2}^2) + exp(-\gamma\mu_{\Delta^2} + \gamma^2\sigma_{\Delta^2}^2/2)^2\right)$$

*where both the expectation and the variance are functions of $\mu_{\Delta^2}$ and $\sigma_{\Delta^2}$.*

*Proof.* $E_\Delta[K(\Delta)]$ and $Var_\Delta[K(\Delta)]$ in Theorem 4.1 are the variance and the expectation of the log-normal distribution. With $\|\Delta\|_2^2 \sim \mathcal{N}$, $exp(-c\|\Delta_i\|_2^2)$ follows the log-normal distribution giving us

$$E[exp(X)] = exp(\mu_x + \sigma_x^2/2), \tag{15}$$

and

$$Var[exp(X)] = [exp(\sigma_x^2) - 1]exp(2\mu_x + \sigma_x^2) \tag{16}$$

for any normal random variable $X \sim \mathcal{N}(\mu_x, \sigma_x^2)$. Thus, with Equation 13,

$$E_{\omega, \Delta}[\cos(\omega^T \Delta)] = E_\Delta[K(\Delta)]$$

$$= E_\Delta[exp(-c\|\Delta\|_2^2)] = exp(-\gamma\mu_{\Delta^2} + \gamma^2\sigma_{\Delta^2}^2/2)$$

and the variance of the log-normal distribution is

$$Var_\Delta[exp(-c\|\Delta\|_2^2)] = [exp(\gamma^2\sigma_{\Delta^2}^2) - 1]exp(-2\gamma\mu_{\Delta^2} + \gamma^2\sigma_{\Delta^2}^2).$$

We have

$$Var_{\omega, \Delta}[\cos(\omega^T \Delta)] = \frac{1}{2} + \frac{1}{2}exp(-2\gamma\mu_{\Delta^2} + 2\gamma^2\sigma_{\Delta^2}^2) + exp(-\gamma\mu_{\Delta^2} + \gamma^2\sigma_{\Delta^2}^2/2)^2.$$

$\square$

**Proposition 4.4.** *For the spherical kernel,*

$$K(\Delta) = 1 - \frac{3}{2}\frac{\|\Delta\|}{\theta} + \frac{1}{2}\left(\frac{\|\Delta\|}{\theta}\right)^3$$

*if $\|\Delta\| < \theta$. 0 otherwise. With $\|\Delta\|_2^2, \|\sqrt{k}\Delta_i\|_2^2 \sim \mathcal{N}(\mu_{\Delta^2}, \sigma_{\Delta^2}^2)$ following the normal distribution for any $i$, the expectation and the variance for the kernel are respectively*

$$E_{\omega, \Delta}\left[\frac{1}{k/2}\sum_{i=1}^{k/2}\cos(\omega_i^T(\sqrt{k}\Delta_i))\right] = E_\Delta[K(\Delta)] = 1 - \frac{3\mu_\Delta}{2\theta} + \frac{\mu_\Delta^3 + 3\mu_\Delta\sigma_\Delta^2}{2\theta^3},$$

$$Var_{\omega, \Delta}\left[\frac{1}{k/2}\sum_{i=1}^{k/2}\cos(\omega_i^T(\sqrt{k}\Delta_i))\right] = \frac{1}{k/2}\left(1 - \frac{9\mu_\Delta}{2\theta} + \frac{9\mu_\Delta^2}{4\theta^2} + \frac{3\mu_\Delta^3 + 9\mu_\Delta\sigma_\Delta^2}{\theta^3} - \frac{3\mu_\Delta^4 + 9\mu_\Delta^2\sigma_\Delta^2}{2\theta^4} + \frac{6\mu_\Delta^4\sigma_\Delta^2 + \mu_\Delta^6 + 9\mu_\Delta^2\sigma_\Delta^4}{4\theta^6}\right)$$

*where the density function $p(\omega_i)$ is the Fourier transform of the kernel $K(\delta)$.*

*Proof.* To use Theorem 4.1, we need to obtain $E[K(\Delta)]$ and $E[K(2\Delta)]$.

$$E_\Delta[K(\Delta)] = 1 - \frac{3E[\|\Delta\|]}{2\theta} + \frac{E[\|\Delta\|^3]}{2\theta^3} \tag{17}$$

$$E_\Delta[K(2\Delta)] = 1 - \frac{3E[\|\Delta\|]}{\theta} + \frac{4E[\|\Delta\|^3]}{\theta^3} \tag{18}$$

We let $\mu_\Delta = E[\|\Delta\|]$ and $\sigma_\Delta = \sqrt{Var[\|\Delta\|]}$. The third non-central moment of a Gaussian is

$$E[\|\Delta\|^3] = \mu_\Delta^3 + 3\mu_\Delta\sigma_\Delta^2 \tag{19}$$

where $\mu_\Delta = \int_\theta^\infty x f_N(x)dx = \int_\theta^\infty x f_{TN}(x)dx \times \int_\theta^\infty f_N(x)dx$ and $\sigma_\Delta = \int_\theta^\infty (x - \mu_\Delta)^2 f_N(x)dx = \int_\theta^\infty (x - \mu_\Delta)^2 f_{TN}(x)dx \times \int_\theta^\infty f_N(x)dx$. $f_N(.)$ is the density function of the normal distribution and $f_{TN}(.)$ is the density function of the truncated normal distribution.

$$E_{\omega, \Delta}[\cos(\omega^T \Delta)] = E_\Delta[K(\Delta)] = 1 - \frac{3\mu_\Delta}{2\theta} + \frac{\mu_\Delta^3 + 3\mu_\Delta\sigma_\Delta^2}{2\theta^3}$$

$$Var_{\omega, \Delta}[\cos(\omega^T \Delta)] = \frac{1}{2} + \frac{1}{2}E_\Delta[K(2\Delta)] + E_\Delta[K(\Delta)]^2 \tag{20}$$

With a little bit of algebra using Equations 17 and 19,

$$E_\Delta[K(\Delta)]^2 = 1 - \frac{3\mu_\Delta}{\theta} + \frac{9\mu_\Delta^2}{4\theta^2} + \frac{\mu_\Delta^3 + 3\mu_\Delta\sigma_\Delta^2}{\theta^3} - \frac{3\mu_\Delta^4 + 9\mu_\Delta^2\sigma_\Delta^2}{2\theta^4} + \frac{6\mu_\Delta^4\sigma_\Delta^2 + \mu_\Delta^6 + 9\mu_\Delta^2\sigma_\Delta^4}{4\theta^6}$$

And, with Equations 18 and 20,

$$Var_{\omega,\Delta}[\cos(\omega^T\Delta)] = 1 - \frac{9\mu_\Delta}{2\theta} + \frac{9\mu_\Delta^2}{4\theta^2} + \frac{3\mu_\Delta^3 + 9\mu_\Delta\sigma_\Delta^2}{\theta^3} - \frac{3\mu_\Delta^4 + 9\mu_\Delta^2\sigma_\Delta^2}{2\theta^4} + \frac{6\mu_\Delta^4\sigma_\Delta^2 + \mu_\Delta^6 + 9\mu_\Delta^2\sigma_\Delta^4}{4\theta^6}$$

$\square$

## A.2 Analysis for The Proposed Fast Random Projection

**Lemma 4.5.** *With $\boldsymbol{C} \in \mathbb{R}^{k \times (D/k)}$, a random matrix with $k \times (D/k) = D$ elements, and each element following the normal distribution $\mathcal{N}(0,1)$ where $D$ is the original dimensionality and $k$ is the dimensionality after projection, the expectation of $\|\boldsymbol{v}\|^2$ is $\mathbb{E}\{\sum_i^k [diag(\boldsymbol{CU})]_i^2\} = \|\boldsymbol{u}\|^2$, and, for each element of $\boldsymbol{v}$, $\frac{(\boldsymbol{v})_j}{\sqrt{\sum_l (\boldsymbol{U})_{l,j}^2}} \sim \mathcal{N}(0,1)$.*

*Proof.*

$$\mathbb{E}\{\sum_i^k [diag(\boldsymbol{CU})]_i^2\}$$

$$= \mathbb{E}\{\sum_i^k [\sum_{j=1}^D (\boldsymbol{C})_{i,j}(\boldsymbol{U})_{j,i}]^2\}$$

$$= \mathbb{E}\{\sum_i^k \sum_{j,j'} (\boldsymbol{C})_{i,j}(\boldsymbol{C})_{i,j'}(\boldsymbol{U})_{j,i}(\boldsymbol{U})_{j',i}\}$$

$$= \mathbb{E}\{\sum_i^k \sum_j (\boldsymbol{C})_{i,j}^2 (\boldsymbol{U})_{j,i}^2\}$$

$$= \sum_i \sum_j (\boldsymbol{U})_{j,i}^2 = \|\boldsymbol{u}\|^2$$

As there are only $D$ non-zero elements in $\boldsymbol{C}$ and $\mathbb{E}\{\sum_i^k [diag(\boldsymbol{CU})]_i^2\} = \|\boldsymbol{u}\|^2$, we have normally distributed

$$\frac{(\boldsymbol{v})_i}{\sqrt{\sum_l (\boldsymbol{U})_{l,i}^2}} \sim \mathcal{N}(0,1)$$

after projection $\boldsymbol{v} = \|diag(\boldsymbol{CU})\|^2$ instead of traditional random projection $\frac{1}{\sqrt{k}}\boldsymbol{Ru}$ with

$$\frac{(\boldsymbol{v})_i}{\sqrt{\|\boldsymbol{u}\|/k}} \sim \mathcal{N}(0,1).$$

$\square$

**Lemma 4.6.** *With probability $1 - 2e^{-(\epsilon^2 - \epsilon^3)k/4}$,*

$$(1-\epsilon)(m/M)\|\boldsymbol{u}_1 - \boldsymbol{u}_2\|^2 \leq \|\boldsymbol{v}_1 - \boldsymbol{v}_2\|^2 \leq (1+\epsilon)(M/m)\|\boldsymbol{u}_1 - \boldsymbol{u}_2\|^2$$

*where $\|\boldsymbol{u}_1\| = \sum_l \sum_m (\boldsymbol{U}_1)_{l,m}^2$ and*

$$m = \min\{\sqrt{\sum_l (\boldsymbol{U})_{l,1}^2}, \sqrt{\sum_l (\boldsymbol{U})_{l,2}^2}, \dots, \sqrt{\sum_l (\boldsymbol{U})_{l,k}^2}\}$$

$$M = \max\{\sqrt{\sum_l (\boldsymbol{U})_{l,1}^2}, \sqrt{\sum_l (\boldsymbol{U})_{l,2}^2}, \ldots, \sqrt{\sum_l (\boldsymbol{U})_{l,k}^2}\}$$

*Proof.* The main difference from the proof of the JL lemma (Vempala, 2004) is that, now in our formulation, we have a generalized chi-square distribution for $\boldsymbol{v}_i$ with

$$\sum_i^k \left( \frac{(\boldsymbol{v})_i}{\sqrt{\sum_l (\boldsymbol{U})_{l,i}^2}} \right)^2$$

instead of the $\chi^2$-distribution $\frac{\|\boldsymbol{v}_1\|^2}{\|\boldsymbol{u}_1\|^2/k} \sim \chi_k^2$ with the JL lemma because the denominator depends on $i$ now. Let us consider the following two inequalities for the $i$-th term of $\|diag(\boldsymbol{CU})\|$:

$$(m/M)k \sum_l (\boldsymbol{U})_{l,i}^2 \le \|\boldsymbol{u}\|^2 \le (M/m)k \sum_l (\boldsymbol{U})_{l,i}^2 \tag{21}$$

$$\frac{(\boldsymbol{v})_i^2}{(m/M)\|\boldsymbol{u}\|^2} \le \frac{(\boldsymbol{v})_i^2}{k \sum_l (\boldsymbol{U})_{l,i}^2} \le \frac{(\boldsymbol{v})_i^2}{(M/m)\|\boldsymbol{u}\|^2} \tag{22}$$

With Inequality 22, one can obtain

$$Pr(\|diag(\boldsymbol{CU})\|^2 > (1+\epsilon)(M/m)\|\boldsymbol{u}\|^2) \le Pr(\chi_k^2 > (1+\epsilon)k)$$
$$Pr(\|diag(\boldsymbol{CU})\|^2 < (1-\epsilon)(m/M)\|\boldsymbol{u}\|^2) \le Pr(\chi_k^2 < (1-\epsilon)k)$$

It is shown in (Vempala, 2004) that

$$Pr(\chi_k^2 > (1+\epsilon)k) = Pr(\chi_k^2 < (1-\epsilon)k) = e^{-(\epsilon^2 - \epsilon^3)k/4}$$

With the union bound, the probability that Inequality 4.6 is satisfied is

$$1 - 2e^{-(\epsilon^2 - \epsilon^3)k/4}$$

$\square$

**Theorem 4.7.** *The expectation and the variance for fast random projection with our method are*

$$E_{\omega,\Delta}[\sum_i (\omega_i^T \Delta_i)^2] = \mu_{\Delta^2}, \quad and \quad Var_{\omega,\Delta}[\sum_i (\omega_i^T \Delta_i)^2] = (2\mu_{\Delta^2} + \sigma_{\Delta^2}^2)/k$$

*where $\|\Delta_i\|_2^2 \sim \mathcal{N}(\mu_{\Delta^2}, \sigma_{\Delta^2}^2)$ and $\omega_i \sim \mathcal{N}(0,1)$.*

*Proof.* From Li et al. (2006a) for fixed $\Delta$, we have

$$E_{\omega}[\sum_i (\omega_i^T \Delta)^2] = \|\Delta\|_2^2$$

Thus, again with the law of total expectation $E_Y[E_X[X|Y]] = E_X[X]$,

$$E_{\omega,\Delta}[\sum_i (\omega_i^T \Delta_i)^2] = E_\Delta[E_\omega[\sum_i (\omega_i^T \Delta_i)^2 | \Delta_i]] = \mu_{\Delta^2}$$

Using the law of total variance $Var_Y(Y) = E_X(Var_Y(Y|X)) + Var_X(E_Y(Y|X))$, we have

$$Var_{\omega,\Delta}[\sum_i (\omega_i^T \Delta_i)^2] = E_\Delta[Var_\omega[\sum_{i=1}^k (\omega_i^T \Delta_i)^2 | \Delta_i]] + Var_\Delta[E_\omega[\sum_{i=1}^k (\omega_i^T \Delta_i)^2 | \Delta_i]]$$

$$= E_\Delta[Var_\omega[\sum_{i=1}^{k}(\omega_i^T\Delta_i)^2|\Delta_i]] + Var_\Delta[\sum_{i=1}^{k}E_{\omega_i}[(\omega_i^T\Delta_i)^2|\Delta_i]]$$

$$= E[\frac{2}{k}\|\Delta_i\|_2^2] + Var[\sum_{i=1}^{k}\|\Delta_i\|_2^2] = \frac{2\mu_{\Delta^2}}{k} + \frac{\sigma_{\Delta^2}^2}{k}$$

$\square$

