# OpenReview forum: "Fast Random Fourier Features Through Randomized Block-Diagonal Projection"
_TMLR — Rejected by TMLR_

### Review · Reviewer_3L9M · 2024-09-08

**Summary Of Contributions:**

This manuscript proposes a method for fast implementation of random projections and the random feature method for kernel approximation. The paper claims a runtime of O(max{k,D} n) where D in the ambient dimension of the datapoints, n is their number and k is the dimension of the projection.  In comparison the runtime is O(knD) in the case of a fully dense random projection matrix, and O(knlog D)  for sparse projections based on the Hadamard-Welsh  and Fast Fourier Transform.

The paper presents the algorithm, attempts a mathematical analysis and presents empirical results where the proposed method appears to be faster than the previously mentioned methods for a comparable accuracy.

**Audience:**

Yes

**Claims And Evidence:**

No

**Requested Changes:**

The paper needs substantial changes for clarity and correctness.

**Strengths And Weaknesses:**

I found several points of concern on the presentation and analysis of the proposed method:
1. First, the theoretical runtime of the Fastfood (FFT-based) method O(k n log D) beats the runtime of the proposed method O(max{k,D}n) when k < D/log D, which is almost the entire interesting range of projection dimensions k. Perhaps the authors can explain better why their method is preferable.
 2. The algorithm is not clearly explained. In particular the matrix C appearing the projection step is not fully explained.
Moreover, the method seems a bit unusual in that an extreme amount of sparsity is used in the projection: there are supposed to be only D non-zero elements in the k x D projection matrix, which leaves D/k non-zero elements per row. In principle a lot of information can be lost.
3. The mathematical analysis is not sound, for instance several quantities which are positive by definition like ell_2 norms of vectors are assumed to be Gaussian random variables. Otherwise it is difficult to follow the thread of the analysis as the notation is not clear and several unjustified approximation are made.

---

> ### Author Response · Authors · 2024-10-29
> **Response to Reviewer 3L9M**
>
> We thank the reviewer for their feedback and suggestions for clarification. We address the concerns below:
>
> > Q1: First, the theoretical runtime of the Fastfood (FFT-based) method O(k n log D) beats the runtime of the proposed method O(max{k,D}n) when k < D/log D, which is almost the entire interesting range of projection dimensions k. Perhaps the authors can explain better why their method is preferable.
>
> For this, we have added further elaborations on the complexity in the sub-section "The Complexity":
>
> "Note also that the big-O notation obtained through asymptotic analysis does not account for constant factors due to the fact that when n gets large enough, constants do not matter. Moreover, it has been suggested that the computational speeds of other related methods like Fastfood relying on the Walsh-Hadamard Transform could be impacted by large big-O constants Iwen et al. (2007), Li et al. (2014).
>
> In our experiments, it is demonstrated that there is a bigger computational advantage with the complexity our linear-time method when both D and k are large shown in Table 3, i.e. there is a 15x speed-up for our method over Fastfood on real-world datasets with larger D and k while the speed-up is only around 3x compared to that of Fastfood with small D and small k. For the value of k, it is shown that empirically, for low errors with random projections and kernel approximations using random Fourier features, k should be larger than or close to $10^4$ (Le et al. (2013), Jacot et al. (2018a), Sutherland & Schneider (2015) Zhang et al. (2019), Nabil (2017))."
>
>
> Apart from the revised sub-sections, please also see our file for extra experimental results in the supplementary material for further detailed explanations on the speedup of our method over Fastfood which is at least 3x with smaller k and smaller D values in the table. The speedups range from 3x to 15x with various values of k and D when our method is compared with Fastfood.
>
>
> > Q2: The algorithm is not clearly explained. In particular the matrix C appearing the projection step is not fully explained.
> Moreover, the method seems a bit unusual in that an extreme amount of sparsity is used in the projection: there are supposed to be only D non-zero elements in the k x D projection matrix, which leaves D/k non-zero elements per row. In principle a lot of information can be lost.
>
> We have re-written the whole paper for more clarity to better explain our method in various sections including the title, the abstract, the introductory section, the sub-section "Contributions", the sub-section "The Complexity" and the section "Our Method".
>
> To sum up, traditional sparse projections lead to information loss when sparsity is moderate.  We propose Randomized Block-Diagonal Projection (RBDP) with structured sparsity in a block-diagonal projection matrix and feature shuffling to retain all information of the original features after projection in projected space with high sparsity and efficiency due to the proposed block-diagonal matrix for projection.
>
> More details on the formulation are now in the revised paper.
>
>
> > Q3: The mathematical analysis is not sound, for instance several quantities which are positive by definition like ell_2 norms of vectors are assumed to be Gaussian random variables. Otherwise it is difficult to follow the thread of the analysis as the notation is not clear and several unjustified approximation are made.
>
> There are two approximations including the kernel approximation using random Fourier features with error bounds provided in the theorems and propositions. The other approximation in Proposition 4.2 is there for the case when bias correction is unnecessary. Bias correction should be used with the unbiased estimator if the condition for approximation is not met, i.e. \mu_{\delta^2}/\sigma_{\delta^2} is close to or less than 1. The proposition has been re-written for more clarity on this. Thank you for letting us know this part should be made clearer.
>
> There is no assumption on the use of Gaussians in the projection matrix and the only assumption here is normality described in Eq. 8 of the revised paper (also pointed out in this feedback from the reviewer) with dependent features justified in Section 4.2 “4.2 The Central Limit Theorem for Weakly Dependent Random Variables”. In that section, the Shapiro-Wilk test is used to determine if the assumption is indeed valid after feature shuffling. As shown in Table 1, with feature shuffling and normalization, the values of $\delta_i$ on all three datasets are normally distributed when $k_{multi} = 200$, i.e. the p-value is higher than 0.05 and the W value is close to 1. Section 4 has been adjusted to clarify this part as well.
>
>
> > "Requested Changes: The paper needs substantial changes for clarity and correctness."
>
> With the suggestions, we have made a substantial effort to re-write the whole paper for clarity on each of the points mentioned.

---

### Review · Reviewer_h6J9 · 2024-09-20

**Summary Of Contributions:**

The authors propose an efficient method to implement Random Fourier Features and Random Projections. The contribution claims to improve over the state-of-the-art methods such as Fastfood and Random Kitchen Sinks pushing the complexity down to $O(max(k,D)n)$ where $D$ is the original dimension of the data, $k$ the dimension of the projection space, and $n$ the number of samples. They validate their claims on different datasets.

**Audience:**

No

**Broader Impact Concerns:**

I do not have any concern due to the theoretical nature of the work.

**Claims And Evidence:**

No

**Requested Changes:**

The manuscript is far from being ready for publication and does not meet the TMLR standards.

**Strengths And Weaknesses:**

I will limit myself in this review to judge only the exposition. Indeed, the level of clarity is not sufficient to understand in a clear way the contributions of this work over the related literature. I will present few strong inconsistency below, but in general the writing does not meet the standard for publication and the authors should strongly revise the manuscript.

I will limit myself in this review to judge only the exposition. Indeed, the level of clarity is not sufficient to understand in a clear way the contributions of this work over the related literature. I will present few strong inconsistency below, but in general the writing does not meet the standard for publication and the authors should strongly revise the manuscript.
- Fastfood introduced without citation in the abstract.
- Link to the code does not exist and is not even a complete link
- Inconsistency in math notation presented. For example, $\chi^2$ distribution sometimes with subscript, sometimes not.
- In section 1,2 (Contributions) the authors mention their proposal: “Our method is motivated by the fact that the order for features do not change distance or similarities between feature vectors as long as the same order is mantained for all feature vectors”. This is completely unclear.
- Fastfood and Random Kitchen Sinks are presented many times over different sections from scratch as if there was no consistency.
- Equation 5 represents the paucity of details given in the mathematical exposition.

---

> ### Author Response · Authors · 2024-10-29
> **Response to Reviewer h6J9**
>
> We thank the reviewer for their feedback and suggestions for the writing style and clarity. We address the concerns below:
>
> > Q1: Fastfood introduced without citation in the abstract.
>
> The citation is now added. In some journals, editors suggest avoiding citations in the abstract but we have now adjusted this. Thanks for the reminder.
>
>
> > Q2: Link to the code does not exist and is not even a complete link
>
> We have now added a footnote in the paper to clarify this: "Due to anonymity for the review, the link to the code repository will be provided after the review process." We followed guidelines at conferences and journals with anonymous reviews on this as we thought that the link to our code could reveal our identities. We will make sure that we include the link once the anonymous review is complete so that the editor can see the code provided.
>
>
> > Q3: Inconsistency in math notation presented. For example, $\chi^2$ distribution sometimes with subscript, sometimes not.
>
> The subscript $k$ in $\chi^2_k$ indicates the degrees of freedom of the $\chi^2$-distribution. We can add $k$ also to the name of the $\chi^2$-distribution making it the $\chi^2_k$-distribution in the paper if that helps. We elaborate on this more with "where $\chi^2_k$ denotes the chi-squared random variable with k degrees of freedom." added to the definition in the revised paper.
>
>
> > Q4: In section 1,2 (Contributions) the authors mention their proposal: “Our method is motivated by the fact that the order for features do not change distance or similarities between feature vectors as long as the same order is maintained for all feature vectors”. This is completely unclear
>
> We have substantially re-written the sections on the motivations and the explanations of the proposed method to make this clearer. Besides, in the section "Contributions", we have removed points less important and we have made our important points clearer:
>
> "The proposed method is a linear-time method. The algorithm is 1.) easy to implement, 2.) with complexity
> O(max{k, D}n), 3.) designed to provide an unbiased estimator having variance inversely proportional to k
> and 4.) shown to achieve a speed-up up to 10,000x over Random Kitchen Sinks and a speed-up up to 15x
> over Fastfood on real-world datasets for the realistic values of D and k."
>
>
> > Q5: Fastfood and Random Kitchen Sinks are presented many times over different sections from scratch as if there was no consistency.
>
> We have re-written various parts to make them more consistent with the sentences and wordings.
>
>
> > Q6: Equation 5 represents the paucity of details given in the mathematical exposition.
> We have added more explanations on Equation 5 to elaborate on it more for clarity in the section "Our Method".
>
> To sum up, traditional sparse projections lead to information loss when sparsity is moderate.  We propose Randomized Block-Diagonal Projection (RBDP) with structured sparsity in a block-diagonal projection matrix and feature shuffling to retain all information of the original features after projection in projected space with high sparsity and efficiency due to the proposed block-diagonal matrix for projection.
>
> We introduce structured sparsity to the projection matrix instead of, randomized sparsity, to retain all information from original features leaving only $D$ non-zero elements in the $k\times D$ projection matrix with sparsity $s=1/k$ which is the fraction of the number of non-zero random numbers generated in the projection matrix. Together with normalization and the shuffling of features, we found that computing random Fourier features and random projections can be highly efficient. Theoretical analysis is provided for the error with encouraging experimental supports.
>
> More specifically, we increase sparsity for more efficient computation to the extent that we can retain all information of the original features in the projected vectors. During projection, each dot product between each row of the projection matrix and the feature vector retains the information of a subset of original features. The resultant $k$-dimensional vector retains all information with the $k$ dot products involving all original features after projection.
>
>
> > "Requested Changes: The manuscript is far from being ready for publication and does not meet the TMLR standards."
>
> With the suggestions, we have made a substantial effort to re-write various sections for clarity including the title, the abstract, the introductory section, the sub-section "Contributions", the sub-section "The Complexity" and the section "Our Method". Please see the revised version of the paper.

---

### Review · Reviewer_j11n · 2024-10-16

**Summary Of Contributions:**

The paper introduces a new method for efficient dimensionality reduction in random projections and random Fourier features by sparsifying the random matrix that the input data is multiplied with. The structured nature of the sparsification ensures that the complexity of the projection operation is reduced from $\mathcal{O}(kDn)$ to $\mathcal{O}(\max\{k,D\}n)$. The properties of the method (kernel approximation and distance preservation) are theoretically the method is evaluated on several real world high dimensional datasets where it gives similar approximation as baselines while providing significant speedup.

**Audience:**

Yes

**Claims And Evidence:**

Yes

**Requested Changes:**

The following issues need to be satisfactorily addressed for me to recommend accepting the paper for publication:

1. I could not find the definition of $\gamma$ in Proposition 4.2. If it has not been defined, please define it clearly.

2. Why is '$c$' in the subscript in the first equation of Proposition 4.3? It is basically calculating the moment-generating function of a gaussian random variable at '$-c$' and so it should be $\exp(-c\mu_{\Delta^2} + \frac{1}{2}c^2\sigma^2_{\Delta^2})$ right? Please check and clarify (also do so for the second equation for the variance).

3. I do not understand how the feature correlation matrices are calculated in Figure 1 after randomly picking two examples. Please provide the exact expressions.

4. I feel that the mean alone is not an adequate measure of the approximation quality in Section 5.2. I would recommend adding plots for P95 error of each method to get a better sense of the overall distribution since there can be applications where we want as few outliers as possible.

5. It seems like BC-TCGA dataset on which the approximation quality is poor is not considered when evaluating the performance of SVMs with the proposed approach in Section 5.3. Please include results with this or another dataset with poor approximation quality even if the results don't look very good since it is important to understand the downstream impact of the drop in approximation quality

**Strengths And Weaknesses:**

Strengths:

1. The proposed approach for efficient dimensionality reduction provides significant speedup over baselines while retaining approximation quality as verified both theoretically and empirically

2. The method is principled, appears theoretically sound and is easy to implement.

Weaknesses:

1. The kernel approximation quality is poor for relatively lower dimensional datasets

---

> ### Author Response · Authors · 2024-10-29
> **Response to Reviewer j11n**
>
> We thank the reviewer for the favourable review and the constructive suggestions. We appreciate the time spent on understanding the technical details. We have revised our paper with the requested changes:
>
> > Requested Change 1: I could not find the definition of $\gamma$ in Proposition 4.2. If it has not been defined, please define it clearly.
> > Requested Change 2: Why is '$c$' in the subscript in the first equation of Proposition 4.3? It is basically calculating the moment-generating function of a gaussian random variable at '$-c$' and so it should be $\exp(-c\mu_{\Delta^2} + \frac{1}{2}c^2\sigma^2_{\Delta^2})$ right? Please check and clarify (also do so for the second equation for the variance).
>
> We have now made the notations in Proposition 4.2 and Proposition 4.3 more consistent for clarity, in the revised paper, we have defined $\gamma$ as the parameter of the Gaussian kernel. $c$ is no longer used and we use $\gamma$ throughout the paper instead as it is just the same parameter. We no longer have $c$ in the subscript with $\mu_{c\Delta^2}$ and $\sigma^2_{c\Delta^2}$ as they can be confusing. $\mu$ and $\sigma^2$ with the subscript in Proposition 4.3 are indeed $c\mu_{\Delta^2}$ and $c^2\sigma^2_{\Delta^2}$. We have re-written the equations for the expectation and the variance. Thank you for the suggestion for us to clarify this. The reason was that the constant $c$ was previously absorbed into the unclear notation with $c$ in the subscript.
>
>
> > Requested Change 3: I do not understand how the feature correlation matrices are calculated in Figure 1 after randomly picking two examples. Please provide the exact expressions.
>
> Indeed, we picked two random examples for the correlations between each pair of features. We will make it more explicit in the paper.
>
>
> > Requested Change 4: I feel that the mean alone is not an adequate measure of the approximation quality in Section 5.2. I would recommend adding plots for
> P95 error of each method to get a better sense of the overall distribution since there can be applications where we want as few outliers as possible.
> > Requested Change 5: It seems like BC-TCGA dataset on which the approximation quality is poor is not considered when evaluating the performance of SVMs with the proposed approach in Section 5.3. Please include results with this or another dataset with poor approximation quality even if the results don't look very good since it is important to understand the downstream impact of the drop in approximation quality
>
> We have done experiments for Change #4 and Change #5. We found that the conclusions are very close to what we found previously. Please see our file for extra experimental results in the supplementary material.
>
>
> > "Weakness: The kernel approximation quality is poor for relatively lower dimensional datasets"
>
> That’s absolutely correct. We aim to deal with expensive computation with potentially large values for n, k and D with big dimensionality, accurate kernel approximations using a lot of random features and large-scale datasets for big data analytics.

---

### Author Response · Authors · 2024-10-29
**Revised Paper and Extra Experimental Results**

We have now posted a substantially revised version of the paper and extra experimental results in the supplementary material to clarify various points with the suggestions from reviewers. We thank all reviewers for their effort on reviewing this work.

---

### Decision · Action_Editor_ch8N · 2024-12-03

**Recommendation:** Reject

**Comment:**

The paper introduces a method for efficient dimensionality reduction using structured sparsity in random projections and random Fourier features for kernel approximation. While the topic is relevant and valuable to the field, the submission does not meet the necessary standards for publication in its current form. Several significant issues hinder its acceptance, including a lack of clarity in the exposition, insufficient mathematical rigor, and limited empirical validation.

The reviewers have provided detailed feedback, highlighting both the strengths and weaknesses of the submission. One reviewer noted that while the method is theoretically interesting and computationally efficient, the empirical evaluation is limited, and the results are inconsistent, particularly for some datasets where the proposed approach underperforms. Another reviewer raised concerns about the algorithm's explanation, particularly regarding the matrix construction and the extreme sparsity used in the projections, which could lead to information loss. This reviewer also pointed out that the mathematical analysis contains unsound approximations and lacks clarity, making it difficult to evaluate the validity of the claims. A third reviewer focused on the lack of clarity in the paper’s writing and mathematical exposition, stating that the manuscript does not currently meet the standards required for publication in TMLR.

In response to the reviews, the authors made substantial revisions, including rewriting sections for clarity, addressing inconsistencies in mathematical notation, and adding experimental results to support their claims. They also clarified some assumptions in their mathematical analysis and attempted to provide more justifications for the proposed method. Despite these efforts, the revisions did not fully resolve the fundamental issues raised during the review process. The theoretical analysis remains challenging to follow, with some approximations and assumptions still appearing unjustified. The empirical evaluation, while expanded, is not comprehensive enough to robustly support the paper’s claims. Furthermore, the lack of consistent and clear exposition significantly diminishes the impact and accessibility of the work.

Overall, while the proposed method has potential, the submission requires significant additional work to address the concerns raised by the reviewers. In its current form, it does not meet the standards expected for publication in TMLR, and I recommend rejecting the paper. The authors are encouraged to refine their mathematical analysis, expand the empirical evaluation to include a broader range of datasets and scenarios, and significantly improve the clarity of the presentation. With these improvements, the paper could make a meaningful contribution to the field in the future.

**Audience:**

While the topic is relevant to TMLR’s audience, the current presentation and evaluation limit the accessibility and potential impact of the findings. The consensus is that the paper does not yet meet the standard required to engage the intended audience effectively.

**Claims And Evidence:**

The consensus among referee is that the claims in the submission are not sufficiently supported by accurate, convincing, and clear evidence. While the proposed method appears promising in theory, significant issues remain in both the clarity of the exposition and the mathematical soundness of the analysis. Furthermore, the empirical evaluation is limited, with mixed results that fail to robustly validate the proposed approach.

**Resubmission Of Major Revision:**

The authors may consider submitting a major revision at a later time.